# *Apodanthera glaziovii* (Cucurbitaceae) Shows Strong Anti-Inflammatory Activity in Murine Models of Acute Inflammation

**DOI:** 10.3390/pharmaceutics16101298

**Published:** 2024-10-04

**Authors:** Maria Lorena de Oliveira Andrade, Pedro Artur Ferreira Marinho, Alisson Macário de Oliveira, Thalisson Amorim de Souza, Samuel Paulo Cibulski, Harley da Silva Alves

**Affiliations:** 1Center of Biological and Health Sciences, Phytochemistry Laboratory, Post-Graduation Program in Pharmaceuticals Sciences, State University of Paraíba, Campina Grande 58429-500, Brazil; maria.lorena.oliveira.andrade@aluno.uepb.edu.br (M.L.d.O.A.); cibulski@servidor.uepb.edu.br (S.P.C.); 2Center of Biological and Health Sciences, Pharmacological Analysis Laboratory, Post-Graduation Program in Pharmaceuticals Sciences, State University of Paraíba, Campina Grande 58429-500, Brazil; pedro.marinho@aluno.uepb.edu.br (P.A.F.M.); alissonmacario@servidor.uepb.edu.br (A.M.d.O.); 3Multi-User Characterization and Analysis Laboratory, Research Institute for Drugs and Medicines (IpeFarM), Federal University of Paraíba, João Pessoa 58059-900, Brazil; thalisson.amorim@ltf.ufpb.br

**Keywords:** Caatinga, saponin, mutagenicity, acute toxicity, anti-inflammatory, phytomedicines, cabeça-de-negro

## Abstract

**Background/objectives:** *Apodanthera glaziovii* is an endemic species from the semi-arid Brazilian, which has limited toxicological and pharmacological studies. This species belongs to a well-studied family known for its bioactive compounds used in treating inflammatory. This study aimed to identify secondary metabolites in the stems from *A. glaziovii*, evaluate toxicity, and investigate the anti-inflammatory potential of the stem hydroalcoholic extract (SHE-Ag). **Methods:** qualitative and quantitative assays were employed to identify secondary metabolites, along with chromatographic analyses and ^1^H and ^13^C NMR. Toxicity was assessed through in vitro hemolytic toxicity, in vivo genotoxicity, and oral acute toxicity tests before the pharmacological assays were conducted. **Results:** phytochemical screening, HPLC and NMR analyses suggested the presence of saponins of the norcucurbitacin class. The SHE-Ag exhibited no hemolytic activity and no mutagenic potential. However, in vivo toxicity at a dose of 2000 mg/kg revealed hematological and biochemical alterations, while the 500 mg/kg dose was safe. In the anti-inflammatory assays, SHE-Ag at 100 mg/kg reduced paw edema by 55.8%, and leukocyte and neutrophil migration by 62% and 68% in the peritonitis model, respectively; inflammatory cell migration by 70% in the air pouch model, outperforming indomethacin, which showed a 54% reduction. **Conclusions**: these findings indicate that SHE-Ag is rich in saponins, confirmed through HPLC and ^1^H and ^13^C NMR analyses. The SHE-Ag also demonstrated low toxicity. The inflammation models used showed a reduction in inflammation, pro-inflammatory cells, and edema, highlighting the significant anti-inflammatory activity of hydroethanolic extract *A. glaziovii* stems.

## 1. Introduction

Anti-inflammatory drugs, widely prescribed and commonly used for the management of pain, inflammation, and fever, encompass a variety of substances, including steroids and non-steroidal anti-inflammatory drugs (NSAIDs) [1]. These drugs can be selective or non-selective for the cyclooxygenase-2 (COX-2) enzyme, which is essential in the inflammatory process. Although effective in reducing inflammation, non-selective COX-2 NSAIDs, in particular, possess additional properties such as analgesia, antipyresis, and antithrombotic effects due to their non-specificity [2,3]. However, this broad therapeutic activity can be associated with adverse effects, including gastrointestinal damage such as peptic ulcers and enteropathy, as well as an increased risk of cardiovascular and renal complications [4].

Selective COX-2 anti-inflammatory drugs, in turn, demonstrate a lower incidence of adverse gastrointestinal effects. However, their cost is considerably higher, and there are concerns related to cardiovascular safety [5]. On the other hand, steroidal anti-inflammatory drugs, known as corticosteroids, despite their anti-inflammatory efficacy, are associated with side effects such as changes in blood pressure and body edema [6]. This issue related to the side effects of available anti-inflammatory drugs underscores the importance of parallel research in search of new alternatives.

Studies investigating natural products (NPs) have been prominent in the search for derivatives with anti-inflammatory potential, aiming to mitigate the side effects associated with commonly used medications [7]. However, despite NPs presenting a variety of promising metabolites for modulating various diseases, many of these compounds remain unknown due to a lack of research focused on isolating these bioactives [8]. An example of this issue is the lack of emphasis on therapeutic discoveries involving plants from the Caatinga biome, exacerbated by the significant degradation of natural resources [9]. This biome, located in the Brazilian Northeast Region, has a semi-arid climate and vast resources for bioprospecting its rich and unique flora [9].

The Cucurbitaceae family stands out for its many native species in the Caatinga, with numerous reports of uses in folk medicine and attracting interest due to their potentially therapeutic metabolites, such as cucurbitacins, which demonstrate anti-inflammatory, antitumor, immunomodulatory, and cardioprotective properties [10]. This family encompasses more than 800 species distributed globally, renowned for its popular edible fruits such as watermelon, pumpkin, cucumber, and melon [11,12].

*A. glaziovii* is botanically and morphologically characterized as a climbing plant with thin, furrowed, and hairy stems. In addition, the stems have simple, thin, hairy tendrils. This monoic species is endemic to the Caatinga biome and is popularly known as “cabeça-de-negro”. Although few studies have explored its therapeutic potential and chemical constituents, anti-inflammatory and antitumor activities have been reported for its genus [13]. Regarding the scarcity of data on the therapeutic potential of *A. glaziovii*, one study was conducted to assess the ability of aqueous extracts from the tubers of *A. villosa* and *A. glaziovii* to reverse the toxic effects of *Bothrops jararaca* venom in in vivo models. However, only the *A. villosa* extract demonstrated a delaying effect on animal mortality [14]. To further expand the knowledge of *A. glaziovii*, the present study aimed to evaluate the secondary metabolites present in this species through qualitative and quantitative phytochemistry assays, high-performance liquid chromatography (HPLC), and ^1^H and ^13^C nuclear magnetic resonance (NMR). Additionally, its toxicity was investigated both in vivo and in vitro, and its anti-inflammatory activity was evaluated in in vivo models. These findings are crucial given the limited number of studies on *A. glaziovii*, which still restricts a deeper understanding of its therapeutic potential.

## 2. Material and Methods

### 2.1. Plant Material and Obtention from A. glaziovii Hidroetanolic Extract

The stems from *A. glaziovii* were collected on 14 January 2023 (dry season) at Fazenda do Ouro (Latitude 8°22′02.2″ S and Longitude 36°24′39.5″ W) municipality of Belo Jardim, a rural area of Pernambuco State within the Caatinga biome (semi-arid, Köppen BSh). A portion of the plant material was separated to create an exsiccate and deposited in the Manuel de Arruda Câmara Herbarium (HACAM), under registration number 2554-HACAM. The project was conducted under the authorization from the National System for Management of Genetic Heritage and Associated Traditional Knowledge (SisGen) with process number AF48253.

The stems from *A. glaziovii* were dried in a forced-air oven at 40 °C for seven days. Subsequently, the dried stems were pulverized using a knife mill and subjected to maceration in ethanol 70% as an extractive solvent to obtain the crude hydroethanolic extract. Solvent exchange was carried out every 72 h over a period of 3 weeks. Afterward, the material was filtered, and the solvent was removed using a rotary evaporator (IKA RV 3 Eco) under reduced pressure at 50 °C. Residual water was removed by lyophilization (JJ Científica LJJ05), resulting in the Stem hydroalcoholic extract from *A. glaziovii* (SHE-Ag), which was stored under refrigeration (4 °C).

### 2.2. Qualitative Phytochemical Characterization

A qualitative phytochemical screening was conducted in SHE-Ag for assessment of alkaloids—using Dragendorff reagent (a solution of potassium bismuth iodide composing of basic bismuth nitrate (Bi(NO_3_)_3_), tartaric acid, and potassium iodide (KI), and when in contact with alkaloids, produces an orange or orange red precipitate), catechins (reaction with hydrochloric acid, changing color to pink to purple, due to formation of phoroglucinol), flavonoids (Shinoda test, in which the SHE-Ag was dissolved in 95% ethanol and, to this solution, a small piece of magnesium foil metal was added; this was followed by 3-5 drops of the concentrated HCl –the intense cherry red color indicates the presence of flavonoids), reducing sugars (Benedict’s test, that contains potassium thiocyanate and forms a copper thiocyanate orange/red precipitate), saponins (stable foam after shake in a graduate cylinder during 15 min), polysaccharides (lugol test, changing the color to indigo indicate the presence of polysaccharides), and tannins (the SHE-Ag was allowed to react with FeCl_3_ and a dark green or deep blue indicate the presence of tannins). All reagents were made in the laboratory itself. All screening tests were performed according to the methodology previously described [15,16].

### 2.3. Determination of Total Phenolics

To determine the polyphenol content present in the SHE-Ag, the classical Folin–Ciocalteau method was applied [17]. Gallic acid (Sigma Aldrich, St. Louis, MO, USA) was used as the standard. Metanolic solutions of the SHE-Ag at concentrations ranging from 5 to 40 μg/mL were mixed with a Folin–Ciocalteau (Sigma Aldrich, St. Louis, MO, USA) reagent in an alkaline medium. The absorbance was measured at 757 nm. The polyphenol concentration was expressed in milligrams of gallic acid equivalents (GAE/mL). Analyses were conducted in triplicate, and the calibration equation for gallic acid was y = 0.0195x − 0.0166 (R^2^ = 0.9979).

### 2.4. Determination of Total Flavonoids

Quantification of flavonoids was conducted following the methodology described by [18], using quercetin (Sigma Aldrich, St. Louis, MO, USA) as standard. Methanolic solutions of SHE-Ag were prepared at concentrations ranging from 1 to 30 μg/mL in the presence of an aluminum chloride solution, and absorbance read at 415 nm. The flavonoid content was expressed in milligrams of quercetin equivalents. All analyses were performed in triplicate, and the calibration equation for quercetin was y = 0.0278x + 0.0145 (R^2^ = 0.9931).

### 2.5. Determination of Total Tannins

Using the methodology established by [19], the SHE-Ag was dissolved in methanol, followed by the addition of a vanillin solution in an acidic medium. Catechin (Sigma Aldrich, St. Louis, MO, USA) served as standard. Readings of both the extract and standard solutions were taken at concentrations from 1 to 30 μg/mL, with absorbances measured at 500 nm. The concentration of condensed tannins was expressed in milligrams of catechin equivalents. All analyses were performed in triplicate, and the calibration equation for catechin was y = 0.0049x + 0.0222 (R^2^ = 0.9985).

### 2.6. ^1^H and ^13^C Nuclear Magnetic Resonance (NMR) and High-Performance Liquid Chromatography (HPLC) Profiles

The NMR analyses were performed on a Bruker Avance spectrophotometer (Bruker, Billerica, MA, USA) operating at 500 MHz for ^1^H NMR and 125 MHz for ^13^C NMR. The SHE-Ag sample was dissolved in dimethyl sulfoxide (DMSO-d_6_) (Cambridge Isotope Laboratories, USA). The characteristic peaks of ^1^H and ^13^C of the solvent used in the analyses served as internal standards during spectrum acquisition and analysis. Chemical shifts (δ) were expressed in parts per million (ppm) and coupling constants (J) in Hertz (Hz).

For the characterization of the chromatographic profile, HPLC was applied using a diode array detector (DAD) SPD-M20A system on a Shimadzu chromatograph (Kyoto, Japan) consisting of two LC-10ADvp high-pressure pumps, an SCL-10Avp controller, a degasser DGU-14a, and an automatic injector with a 20 µL sample loop. Chromatographic analyses were conducted using an ACE C18 chromatographic column with a particle size of 5 μm and dimensions of 250 × 4.6 mm. The elution system consisted of ultrapure water acidified with 0.1% formic acid (Sigma Aldrich, St. Louis, MO, USA) (Solvent A) and methanol (Merck, Darmstadt, Germany) (Solvent B) with a linear gradient from 5% to 95% of B over 60 min. The flow rate was maintained constant at 0.6 mL/min. UV spectra were monitored at wavelengths ranging from 190 to 800 nm. The SHE-Ag was analyzed at a concentration of 1 mg/mL. All solutions were filtered through 0.45 μm PVDF membranes with a diameter of 30 mm (Allcrom, São Paulo, Brazil) before use, and all analyses were performed in triplicate.

### 2.7. Assay of Hemolytic Activity

The hemolytic effect of SHE-Ag was measured according to the methodologies previously described [20]. In summary, the experimental procedure involved mice blood collected in tubes with ethylenediamine tetraacetic acid (EDTA). Plasma was separated after centrifugation and washed three times with saline solution. The obtained red blood cells (RBC) were diluted in saline solution (to 5% concentration), which was added to test tubes containing 1.0 mL of SHE-Ag solutions at concentrations of 2000, 1000, 500, 250, and 125 μg/mL. After 1 h, the test tubes were centrifuged, and the supernatant was evaluated using a Shimadzu UV-1900 spectrophotometer at a wavelength of 540 nm. Triton X-100 (Sigma Aldrich) at 1% was used as 100% hemolysis standard and saline as blank (0% of hemolysis). The analysis was performed in triplicate, and the calculation of the hemolytic potential (HP) of the substances was determined using the following equation: HP = (Ae − Ab)/At × 100 when HP = Hemolytic Potential (as a percentage); Ae = Absorbance of the extract-treated RBC; Ab = Absorbance of the blank; At = Absorbance of Triton X-100-treated RBC.

### 2.8. Animals

Female Swiss mice (25–30 g) aged 8 to 10 weeks, acquired from the Animal Facility of the Federal University of Pernambuco, were kept under controlled temperature conditions (22 °C ± 1 °C), with a 12 h light/dark cycle and access to water and standard diet (ad libitum). Following the experiments, all animals were euthanized with an overdose of ketamine (Cristalia, São Paulo, Brazil) combined with xylazine (Syntec, São Paulo, Brazil). Initially, ketamine at 100 mg/kg and xylazine at 10 mg/kg were used to induce sedation, followed by ketamine at 300 mg/kg and xylazine at 30 mg/kg to achieve deep sedation and euthanasia to minimize animal suffering. The experimental protocol was approved by the Ethics Committee on Animal Experimentation of the Federal University of Pernambuco under protocol number 0102/2023, in accordance with the guidelines of the National Council for Animal Experimentation Control (CONCEA).

### 2.9. Acute Oral Toxicity

To determine the toxicity of SHE-Ag, this assay was performed following the guidelines of The Organization for Economic Cooperation and Development [21], with some modifications. Briefly, the animals were divided into three groups (n = 3): a control group that received saline solution (vehicle) and an experimental group that received 2000 or 500 mg/kg of SHE-Ag orally. The mice were initially observed for 6 h and then daily for 14 days for signs of consciousness and disposition, motor activity and coordination, reflexes, and central nervous system activities.

In addition, to ascertain the safety of SHE-Ag for administration in subsequent tests, an assessment of the biochemical and hematological parameters of the mice was conducted. At the end of the toxicity test, blood samples were collected, and the following biochemical parameters were evaluated: total protein, albumin, alanine aminotransferase (ALT), aspartate aminotransferase (AST), alkaline phosphatase, gamma-glutamyl transferase (GGT), urea, and creatinine. Specific kits for each parameter were used (Labtest Diagnóstica, Lagoa Santa, Brazil) within a COBAS Mira Plus analyzer (Roche Diagnostics Systems, Basel, Switzerland). Hematological analysis was performed using an automated hematology analyzer (ABC Vet Animal Blood Counter, Montpellier, France) and optical microscopy. The parameters evaluated included the number of red blood cells, hemoglobin, hematocrit, mean corpuscular volume (MCV), mean corpuscular hemoglobin (MCH), mean corpuscular hemoglobin concentration (MCHC), and total and differential leukocyte counts.

### 2.10. Mammalian Erythrocyte Micronucleus Test

To evaluate the mutagenic profile of the treatment with *A. glaziovii* extract, the micronucleus test in mouse erythrocytes was used based on the method previously described [22], with slight modifications. Briefly, groups of male mice (n = 5) were treated with PBS, cyclophosphamide (50 mg/kg) (Sigma Aldrich), or SHE-Ag (2000 mg/kg) orally in a single dose. At 24 and 48 h after treatment, 10 µL of peripheral blood (collected from the tail vein) were applied to slides, stained with acridine orange, and analyzed under a fluorescence microscope. A total of three slides per animal were prepared, and the micronucleus frequency was determined after counting 2000 erythrocytes per slide [23].

### 2.11. Carrageenan-Induced Paw Edema Model of Acute Inflammation

This test aims to observe the influence of SHE-Ag in reducing some signs of inflammation, such as edema [24]. Male mice were grouped into five groups (n = 6) and treated with SHE-Ag (25, 50, or 100 mg/kg), saline solution, or dexamethasone (10 mg/kg i.p.) (Sigma Aldrich) 30 min before the tests began. After the treatment, the paw edema was induced with carrageenan 2% (15 μL) injected into the plantar region of the right paw. The animals had the volume of their hind paws measured using a digital caliper. Paw size was evaluated at 1, 2, 3, and 4 h.

### 2.12. Peritonitis Model of Acute Inflammation

Seeking to verify the anti-inflammatory activity of SHE-Ag, the evaluation of peritoneal exudate from mice after exposure to the inflammatory agent was performed. Following the method described by [25,26], animals developed acute inflammation after injection of carrageenan (1%). Male mice were grouped into five groups (n = 6) and pretreated with SHE-Ag (25, 50, or 100 mg/kg), saline solution, or dexamethasone (10 mg/kg i.p.) 30 min before the test. After 4 h, the animals were euthanized, and 2 mL of heparinized PBS was injected into the peritoneal cavity. The leukocyte count from peritoneal lavage was performed using an automated hematological analyzer and expressed as a percentage.

### 2.13. Carrageenan-Induced Air Pouch Model of Acute Inflammation

The air pouch model is a widely used inflammation model for screening drugs with potential anti-inflammatory action [27]. Animals received subcutaneous injections of 2.5 mL of sterile air (on days 1 and 4) in the dorsal region. On day 7, 1 mL of carrageenan solution 1% (*w*/*v*) was injected into the pouch cavity. SHE-Ag (25, 50, or 100 mg/kg), the standard drug (indomethacin 10 mg/kg i.p) (Sigma Aldrich), and the vehicle (saline solution) were administered orally 30 min prior to carrageenan injection. After 6 h, total leukocyte counts in the pouch were performed using an automated hematological analyzer [28,29].

### 2.14. Statistical Analysis

The data obtained were analyzed using GraphPad Prism^®^ version 8.0 and expressed as mean values with standard deviation (±SD) or standard error of the mean (±SEM). Statistically significant differences were calculated using one-way analysis of variance (ANOVA), followed by Bonferroni’s or Tukey’s tests (when necessary). Values were considered significantly different when the *p*-value was less than or equal to 0.05.

## 3. Results

### 3.1. Phytochemical Profile of SHE-Ag

The phytochemical screening tests conducted on SHE-Ag did not show the presence of alkaloids, tannins, flavonoids, catechins, reducing sugars, and polysaccharides. However, the saponin test showed a high foam formation. This intense foam was persistent (>15 min), suggesting the presence of saponins. It is important to consider that the phytochemical screening tests conducted may have limitations in sensitivity to detect metabolites in very low concentrations.

As phenolic compounds are ubiquitously distributed in plants, we quantitatively determine these compounds in SHE-Ag using the standard Folin-Ciocalteau method. The polyphenol concentration SHE-Ag found is 34.12 µg GAE/mL, representing 3.4% of the gallic acid equivalent concentration. Regarding tannins and flavonoids, although qualitative assays have low sensitivity, they were consistent with the results of the quantitative assays, which were not detected in SHE-Ag.

### 3.2. ^1^H and ^13^C Nuclear Magnetic Resonance (NMR) and High-Performance Liquid Chromatography (HPLC) Profile

The main peaks identified in the chromatogram (Figure 1) were numbered from 1 to 6. Peak 1, with retention time (Rt) of 7,2 min and absorption bands of 233 and 254 nm, this high polarity substance is suggestive of a phenolic compound, which corresponds to this wavelength. The peaks 2 and 3, with Rt of 48.8; 51.4 min; and the peaks 4, 5, and 6, correspond to the Rt 53.5; 55.4; 64.9 min, respectively, exhibited similar absorption spectra in the PDA, with absorption peaks of 235, 275 and 315 nm. This absorption profile is indicative of norcucurbitacins, also known as cayaponosides, which are triterpenoid saponins featuring an aromatic A-ring [30].

The analysis of HPLC-DAD revealed the absence of peaks/substances with classical absorption profiles, such as alkaloids and flavonoids. Therefore, the HPLC profile is consistent with the qualitative phytochemical screenings, as no peaks (with characteristic absorbance) were detected at Rt corresponding to the quercetin, gallic acid, and catechin, standards for flavonoids, polyphenols, and tannins, respectively.

In the ^1^H NMR spectrum (DMSO, 500 MHz) (Appendix A), signals between *δ*_H_ 6.84 and 5.67 were observed, suggesting hydrogen atoms attached to sp^2^ carbons, with a peak at *δ*_H_ 6.41 (s, ^1^H) characteristic of aromatic hydrogen present in norcucurbitacin-type structures [31]. Additionally, an envelope of signals between *δ*_H_ 3.06 and 1.00 was observed, suggestive of methylene and methyl groups, which reinforces the presence of a triterpenoid nucleus. Furthermore, peaks between *δ*_H_ 3.83 and 3.06 indicate the presence of glycosidic units in the extract, along with signals at *δ*_H_ 4.90 (m) and *δ*_H_ 4.26 (d) suggestive of anomeric protons, characteristic of the presence of sugars in the molecules. This suggests the presence of saponins as major compounds in the spectrum [30,32].

In the ^13^C NMR spectrum (DMSO, 500 MHz) (Appendix A), peaks were observed between *δ*_C_ 144 and 125 ppm, indicative of aromatic carbons from norcucurbitacin nuclei [31]. Signals between *δ*_C_ 53 and 42 ppm can be attributed to methine and methylene carbons in triterpenoid structures, and signals between δ_C_ 30 and 11 ppm are characteristic of methyl groups in the cucurbitane nucleus. Peaks suggestive of oxymethine carbons and anomeric carbons, observed from *δ*_C_ 76 to 61 ppm and at *δ*_C_ 104 and 102 ppm, respectively, indicate the presence of sugars in the spectrum. These findings suggest that the major compounds in the extract are triterpenoid saponins [33].

### 3.3. Hemolytic Activity

The hemolytic potential of SHE-Ag was determined using the hemolytic assay, commonly employed to assess whether the plant extract interacts with erythrocyte membranes and induces alterations, such as their rupture. SHE-Ag did not induce significant hemolysis at the tested concentrations, with the hemolytic potential of the highest concentration (2000 μg/mL) corresponding to 4.8%. The lower concentrations tested did not show hemolysis.

### 3.4. Acute Oral Toxicity Test

Intoxications from raw plant materials are common, especially with an unknown plant species. Therefore, to determine the toxicity level of SHE-Ag, an acute oral toxicity assay was conducted. It was observed that oral administration at a dose of 2000 mg/kg resulted in stimulatory behavioral changes, such as increased locomotor activity and grooming movements in the first 30 min, which ceased after 60 min. Additionally, water and food consumption decreased, differing significantly from the saline-treated animals (*p* < 0.05) (Table 1).

Although no deaths were recorded at any of the tested doses, due to the observations of biochemical and hematological parameters (discussed below) and the changes in water and food consumption, an additional toxicity test was conducted using a dose of 500 mg/kg, which are five times higher than the doses tested in pharmacological assays. The dose of 500 mg/kg did not cause behavioral changes or signs of toxicity during the 14-day observation period. Both evaluated doses (2000 and 500 mg/kg) did not affect the relative organ weight nor exhibit macroscopic alterations in the internal organs.

In the analysis of biochemical parameters, no changes were observed in the ALB, ALT, AST, TP, ALP, GGT, Urea, CREAT, TC, and TG values at 500 mg/kg dose (Table 2). However, at the 2000 mg/kg dose, the hepatic markers ALB, ALT, and AST were significantly increased (Table 2). Additionally, no alterations were observed in the renal function markers Urea e CREAT, nor in the concentrations of triglycerides and total cholesterol at either dose.

Regarding hematological parameters, a reduction in hematocrit, hemoglobin, and MCH values was observed in animals treated with the 2000 mg/kg dose (Table 3). However, no hematological changes were observed in animals treated with the 500 mg/kg dose (Table 3). The values for MCV, MCHC, leukocytes, segmented cells, lymphocytes, and monocytes did not show significant differences at either dose evaluated (Table 3).

### 3.5. DNA Damage Results by Micronucleus Test

The objective of this test was to determine whether SHE-Ag could induce DNA damage, as the presence of micronuclei indicates a mutagenic profile in the tested sample [34,35]. The evaluated results showed that treatment with SHE-Ag did not induce DNA alterations at the tested concentration, presenting values similar to the saline-treated group (Table 4).

### 3.6. Carrageenan-Induced Paw Edema

To evaluate the anti-inflammatory activity of SHE-Ag in vivo, the reduction in paw edema was investigated using a carrageenan-induced paw edema model in mice. Tissue swelling was observed following carrageenan administration. However, the groups treated with SHE-Ag exhibited a reduction in swelling at the tested concentrations of 25 mg/kg (34.4%, *p* < 0.05), 50 mg/kg (48.8%, *p* < 0.05), and 100 mg/kg (55.8%, *p* < 0.05), demonstrating a dose-dependent concentration (Figure 2). All SHE-Ag treated groups showed significantly low paw edema (*p* < 0.05) when compared to the group receiving carrageenan 4 h after its administration.

### 3.7. Peritonitis Test

In this other test, the induced peritoneal inflammation method using carrageenan was employed, which causes increased abdominal vascular permeability after 4 h of exposure to the inflammatory agent. Upon analysis of post-carrageenan peritoneal exudates, we observed that the total number of leukocytes treated with vehicle (saline) was 5.26 ± 0.28 × 10^5^. On the other hand, in the dexamethasone-treated mice (10 mg/kg), the leukocyte migration was inhibited in 60.3% when compared to the vehicle group, primarily due to neutrophil influx, corresponding to 65.7% compared to the vehicle. Treatment with SHE-Ag (25, 50, or 100 mg/kg) prior to the inflammatory agent resulted in a significant reduction in leukocyte counts (Figure 3). This significant reduction was observed in mice treated with all doses: 25 mg/kg (21.9%, *p* < 0.05), 50 mg/kg (40.3%, *p* < 0.05), and 100 mg/kg (62%, *p* < 0.05).

In this perspective, we observed that the greatest inhibition of leukocyte migration into the peritoneal cavity among mice was observed due to neutrophil infiltration at the dosage of 100 mg/kg (68.9%, *p* < 0.05). In addition, the data demonstrated that the reduction in carrageenan-induced neutrophil infiltration occurs in a dose-dependent concentration.

### 3.8. Carrageenan-Induced Acute Inflammation in Air Pouch Model

To complement the evaluation of the anti-inflammatory activity of SHE-Ag, the air pouch test was conducted. In this model, SHE-Ag demonstrated a significant reduction in leukocyte migration at all tested dosages, with decreases of 25.4%, 52.7%, and 70.5% for the doses of 25, 50, and 100 mg/kg, respectively (Figure 4). In comparison, the indomethacin-treated group (10 mg/kg) showed a 54% inhibition. These results indicate a dose-dependent response, with the most pronounced effect observed at the 100 mg/kg dosage when compared to the saline-treated group.

## 4. Discussion

Research on NPs for therapeutic purposes presents a promising alternative for reducing undesirable side effects [7]. However, due to the vast diversity of plant species, coupled with the complexity of metabolomic studies, many of which remain undiscovered. Although the Cucurbitaceae family has been “extensively” studied, especially genera such as *Citrullus*, *Cucumis*, *Cucurbita*, and *Momordica*, with known and edible species such as watermelon (*Citrullus lanatus*), cucumber (*Cucumis sativus*), pumpkin (*Cucurbita pepo*) and melon (*Momordica charantia*), *Apodanthera* genus and others are less addressed in the scientific literature [36].

The Brazilian semi-arid (Caatinga), defined by low rainfall rates (<800 mm/year), is classified by a hot semi-arid climate (BSh) in the Köppen climate classification, with high temperatures and great seasonality in precipitation. The Caatinga flora is characterized by a xerophytic formation, a form of seasonally dry low forest. The vegetation cover is sparse, spreading across the massifs and plateaus through which the rivers flow, generally intermittently [37]. Despite the arid scenario, a large variety of plant families were well adapted to this climate, such as Fabaceae, Euphorbiaceae, Cactaceae, Asteraceae, Malpighiaceae, and Cucurbitaceae. Regarding Cucurbitaceae, from the nine species of the hemicryptophytic *Apodanthera* genus present in Brazil, five were found in the Caatinga biome, with *A. glaziovii* restricted this Biome, described in Alagoas, Bahia, Paraíba, Pernambuco, and Sergipe States [38].

The Cucurbitaceae family plays a significant role in ethnopharmacology due to the diversity of species and unique bioactive compounds. Extracts and compounds isolated from these plants have been studied for their medicinal properties, which include antioxidant, anti-inflammatory, antimicrobial, and antitumor activities. The congener *A. glaziovii* species, *A. congestiflora,* has been described in traditional/folk medicine for controlling general, including dental pain. In the veterinary field, the use of the tuber of this species is reported for prophylaxis and treatment of Newcastle disease in birds [39]. Furthermore, *A. congestiflora* is used as a “blood purifier”, and baths containing the plant were used against skin blemishes [40].

In this study, aiming to identify classes of bioactive compounds in *A. glaziovii* stems hydroethanolic extract, the presence of saponins was observed in qualitative foam test, along with a low concentration of phenolic compounds, and the absence of flavonoids and tannins using the employed methodologies. A striking characteristic of species in the Cucurbitaceae family is the high quantity of succulent sap rich in saponins, which some species have developed as an adaptation to water scarcity, potentially explaining the elevated saponin content in the stems of these plants [41]. One of the main classes of secondary metabolites found in Cucurbitaceae species is cucurbitacins—a highly oxygenated triterpenic saponins with a myriad of therapeutic potentials [36]. In this context, the presence of saponins was observed in qualitative assays (as the main metabolite), consistent with the nature of metabolites found in other species of this family/genus.

Interestingly, a related species of the same genus, *A. congestiflora*, also showed similar results. In the ethanolic extract of its tuberous roots, this species exhibited low levels of total phenolic compounds and was characterized by the predominant presence of cucurbitacin-type saponins [42]. It is important to consider that, in contrast to the results obtained for *A. glaziovii*, *A. congestiflora* showed a high content of flavonoids. This discrepancy in results may be associated with environmental conditions, especially the water scarcity faced by the species, which can influence the presence, absence, and content of these chemical compounds [43].

In the HPLC analyses conducted with SHE-Ag, it was possible to observe absorption peaks at 280, 257, and 275 nm, which are characteristic of cucurbitacins and norcucurbitacins, such as Cayaponoside A1, which have a strong absorption band at 300 nm [44]. Furthermore, the analysis of ^1^H and ^13^C NMR spectra corroborated that these saponins are the major compounds present in this plant species. These saponins are known for their cytotoxic potential, deeply studied in the research of antitumoral compounds, as well as their well-documented chemoprotective and anti-inflammatory potential [45].

Thus, due to the cytotoxic action described for saponins, the hemolytic evaluation of SHE-Ag was initially carried out using mice RBC. It was observed that SHE-Ag did not damage RBC. Studies of the Cucurbitaceae family and the *Apodanthera* genus, using similar assays to determine hemolytic activity, also reported the absence of membrane damage [13,42,46]. As defined by [47], hemolysis indices lower than 10% indicate security to experimental models. So, once SHE-Ag did not induce significant hemolysis at the high concentration tested (2000 mg/mL), with a hemolytic index of 4.8%, it can be considered safe for administration in animal models.

For a long time, the cytotoxic and hemolytic potential of saponins has been attributed to their amphiphilic chemical nature, characterized by a hydrophobic component in the aglycone portion (triterpenoid) and a hydrophilic portion related to the glycosidic structure [48]. However, the more polar portion of saponins directly influences their hemolytic and cytotoxic capabilities. Parameters such as interglycosidic linkages, types of sugar units, and substitution patterns can influence both the cytotoxic and hemolytic activities of saponins [49]. Modifications in the aglycone structure also play a crucial role in the hemolytic and cytotoxic activity of saponins. Studies with synthetic derivatives of Pulsatilla saponin D revealed that the double bond in the C ring, especially between carbons C-12 and C-13, was essential for these activities. Furthermore, it was observed that increased lipophilicity was correlated with an increase in cytotoxic activity. These studies identified a derivative with this double bond between carbons C-12 and C-13 that was non-hemolytic but demonstrated cytotoxic potential [50].

For a long time, it was believed that cytotoxicity was closely linked to hemolytic potential. However, new research suggests that there may be variations in these characteristics, affecting only one of the activities. For example, some saponins may exhibit cytotoxic activity without causing hemolytic toxicity, while others may be hemolytic without demonstrating cytotoxicity. This diversity of results is significant, especially considering the potential anticancer properties of saponins without the associated hemolytic toxicity [49,50].

Considering the structure of cucurbitacins and cayaponosides, which may be present in SHE-Ag, and highlighting the data from preliminary phytochemical assays, HPLC-DAD absorbances, and the analysis of ^1^H and ^13^C NMR spectra, it is observed that the C ring generally lacks the double bond associated with cytotoxic and hemolytic activity. These characteristics may explain the absence of hemolytic activity in SHE-Ag.

The evaluation of the toxicity of a substance or plant extract is a complex parameter that involves more than just the mortality of the animal. Therefore, signs of alterations in hepatic, renal, cardiac, and other activities must be assessed [51]. This study determined, by observing the hepatic and hematological parameters of the animals, that the dosage of 2000 mg/kg showed significant alterations, indicating toxicity. Considering the absence of the hemolytic potential of SHE-Ag, it can be inferred that the hematological toxic effects identified in the in vivo assay are not related to alterations in erythrocyte membranes. These alterations need to be further investigated regarding the myelotoxic potential of SHE-Ag, as this convergence of results suggests a possible alteration in the production of cells in the hematopoietic system. Regarding the mutagenicity test, it was found that there was no alteration in the genetic material, as no micronuclei formations were observed in the peripheral blood of the treated animals.

Since the liver is one of the primary organs for metabolizing chemical substances, the chemical components present in *A. congestiflora* and *A. glaziovii* could potentially involve the hepatic metabolic pathway [52]. Further research is necessary to identify the pathways of toxicity for these altered metabolic functions and to investigate other potential alterations. Considering the described parameters and the OECD guidelines [21], substances showing toxicity > 300–2000 mg/kg are classified into “category 4”. The second tested dosage in acute toxicity test (500 mg/kg), which range between 300 and 2000 mg/kg, showed no signs of toxicity and was proved safe. Following the confirmation of the SHE-Ag’s lack of toxicity at the dosage of 500 mg/kg, further in vivo assays were conducted, aiming to explore new alternatives to address the issues associated with currently available anti-inflammatories. While there are options that address the limitations of non-selective NSAIDs, their cost and adverse cardiovascular events remain concerning.

Anti-inflammatory potential is defined by the ability of a substance to reduce the recruitment of pro-inflammatory cells and the exudate originating from inflammation [53]. The inflammatory process recruits immune system cells such as macrophages, neutrophils, and specialized cells like cytokines, prostaglandins, leukotrienes, and pro-inflammatory cytokines (TNF-α, IL-1β, IL-6) [54,55]. Thus, substances like carrageenan can modulate an inflammatory response by recruiting these cells and promoting inflammation. Drug assays evaluating anti-inflammatory potential commonly use carrageenan, which increases inflammation mediators one hour after exposure, with potential peaks at various subsequent times [54].

The assays conducted in this study employed carrageenan to evaluate the anti-inflammatory activity of SHE-Ag. In the paw edema test, conducted at doses of 25, 50, and 100 mg/kg, previously established based on acute toxicity assay results, a significant reduction in edema was observed. This reduction can be attributed to various mechanisms, such as inhibition of pro-inflammatory mediator release, suppression of immune response, or modulation of inflammatory pathways. However, further pharmacokinetic studies are necessary to elucidate the mechanism of action of SHE-Ag.

Pharmacological studies on saponins suggest that a significant part of their mechanism of action involves reducing immune responses, primarily by modulating inflammatory cytokines through pathways such as NF-κB, TLR4, and MAPKs [56]. In a study with cucurbitacin E (CuE) conducted on IL-1β-induced human chondroblasts, a dose-dependent reduction in expression of IL-1β-induced factors was observed [57]. These findings, combined with the anti-inflammatory data observed in the paw edema model, suggest potential pathways through which cucurbitacins present in SHE-Ag may exert their activity.

The results of the carrageenan-induced peritonitis test demonstrated a significative anti-inflammatory activity in the three tested doses of SHE-Ag. This test evaluates the recruitment of inflammatory cells. A significant reduction of 68% in neutrophils and 62% in leukocytes was observed at the dosage of 100 mg/kg. In comparison, the group treated with dexamethasone showed reductions of 65% and 60% in neutrophils and leukocytes, respectively.

The saponins present in SHE-Ag may act as immunosuppressants, following a pathway similar to dexamethasone. These metabolites can inhibit the release of neutrophils, as their activation is mediated by pro-inflammatory cytokines such as TNF-α and IL-1β. This mechanism has been observed in saponins such as cucurbitacins, including cucurbitacin E (CuE), cucurbitacin B (CuB), cucurbitacin R (CuR), and cucurbitacin IIa (CuIIa), found in the Cucurbitaceae family [57,58,59,60].

CuB, the main compound identified in the methanolic extract of *Cucumis prophetarum* fruits (Cucurbitaceae), demonstrated anti-inflammatory activity in a carrageenan-induced prostatic inflammation model in rats. This activity was attributed to the reduction in the expression of inflammatory mediators such as TNF-α, IL-1β, COX-2, and iNOS, along with inhibition of neutrophil infiltration [58]. In another study, CuB was capable of reducing the secretion of pro-inflammatory cytokines such as IL-1β, IL-6, and TNF-α, as well as nitric oxide (NO) and prostaglandin E2 (PGE2), by lipopolysaccharide (LPS)-stimulated macrophages via TLR4 receptors [61].

Considering the generalized inflammatory response observed in the peritonitis model, the air pouch assay was implemented to validate the results. In this model, the inflammatory process is more focused, controlled, and restricted to the site of substance administration [62]. It allows for the evaluation of inflammatory exudates and the cell count of inflammatory mediators. The results indicated that SHE-Ag showed a significant reduction in leukocyte migration at all tested doses, with a 70% reduction in inflammatory mediator migration at the dose of 100 mg/kg, while dexamethasone demonstrated a reduction of 54%.

Although SHE-Ag showed strong anti-inflammatory potential in the conducted assays, it is important to note that each test evaluates specific parameters of inflammation. In the case of the air pouch assay, which is a more specialized pathway, SHE-Ag showed more promising results. This response pattern can be compared to the mechanism of action observed for cucurbitacin R (CuR) in an experimental model of arthritis induced in rats, which represents a localized and centralized inflammation. In this model, CuR demonstrated efficacy in reducing nitrite, PGE2, and TNF-α levels, as well as reducing the number of inflammatory cells, without affecting cyclooxygenases [59].

In another study, CuR isolated from the roots of *Cayaponia tayuya* (Cucurbitaceae) demonstrated anti-inflammatory activity by reducing the number of inflammatory cells [63]. It is relevant to mention that cayaponosides, initially isolated from *C. tayuya*, were also found in the species *A. congestiflora* [13], exhibiting UV absorption patterns similar to those observed in SHE-Ag HPLC profile, and ^1^H and ^13^C signals in NMR spectra. These findings highlight the potential of triterpenic saponins, such as cucurbitacins, as potent anti-inflammatory molecules, possibly through modulation of the secretion of pro-inflammatory cytokines such as IL-1β, IL-6, and TNF-α [57,59,61].

## 5. Conclusion Remarks

In the phytochemical analyses conducted with SHE-Ag, a low concentration of phenolic compounds and the presence of triterpenic saponins, cucurbitacin-type, were observed. This was confirmed through data obtained from the HPLC fingerprint and the ^1^H and ^13^C NMR data. In in vitro toxicity tests, no hemolytic activity was observed. However, in in vivo toxicity assays, some biochemical and hematological parameters were altered at 2000 mg/kg SHE-Ag dose, though no toxicity was indicated when the dose was reduced to 500 mg/kg. No mutagenic activity was observed. Regarding pharmacological activity, SHE-Ag demonstrated strong anti-inflammatory effects in three models of acute inflammation: paw edema, peritonitis, and air pouch. Therefore, SHE-Ag showed low toxicity and efficacy in reducing the inflammatory process, possibly due to the modulation of pro-inflammatory cytokine secretion, such as IL-1β, IL-6, and TNF-α, as supported by bibliographic evidence. These data justify more advanced studies with this plant species to map these secondary metabolites responsible for the anti-inflammatory activity and to evaluate the mechanisms of action, with the aim of developing new phytomedicines.

## Figures and Tables

**Figure 1 pharmaceutics-16-01298-f001:**
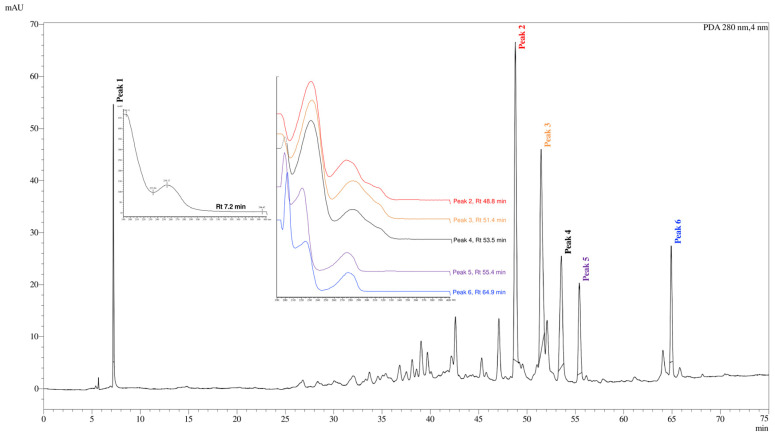
Stem hydroalcoholic extract from *A. glaziovii* (SHE-Ag)’s HPLC profile. RP-HPLC chromatogram of SHE-Ag crude extract at λ 280 nm. In detail, the UV–absorption spectra of the major peaks.

**Figure 2 pharmaceutics-16-01298-f002:**
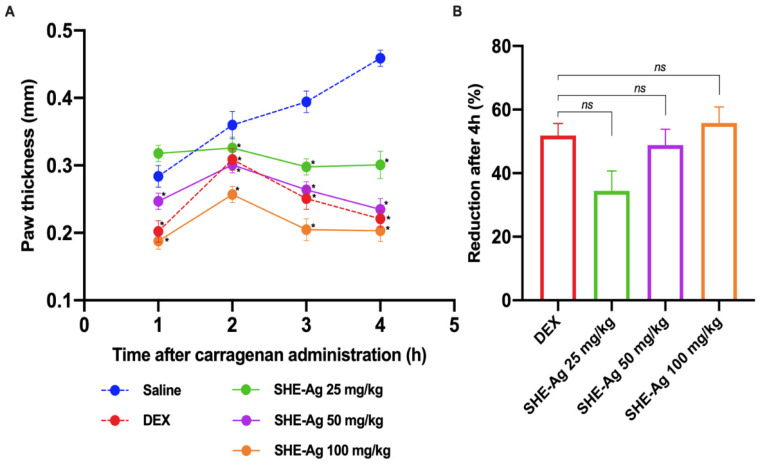
Effect of Stem Hydroalcoholic Extract from *A. glaziovii* on paw edema induced by carra geenan in mice. (**A**) Paw edema was measured hourly for up to 4 h using a caliper. (**B**) The percentage of edema reduction after 4 h of carrageenan administration. The saline-treated group was used as a control group. All results were expressed as mean ± standard error of the mean (SEM). ns, non-significative. * *p* < 0.05.

**Figure 3 pharmaceutics-16-01298-f003:**
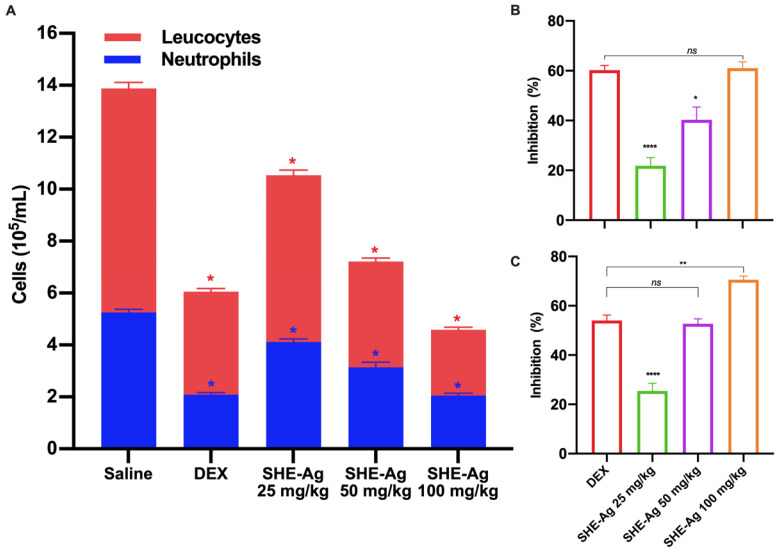
The Effect of Stem Hydroalcoholic Extract from *A. glaziovii* on carrageenan-induced peritonitis in mice. (**A**) Total leukocytes and neutrophils into peritoneal lavage. (**B**) Percentage of inhibition of leukocytes. (**C**) Percentage of inhibition of neutrophils. All results were expressed as mean ± standard error of the mean (SEM). ns, non-significative. **p* < 0.05, ** *p* < 0.01, **** *p* < 0.0001.

**Figure 4 pharmaceutics-16-01298-f004:**
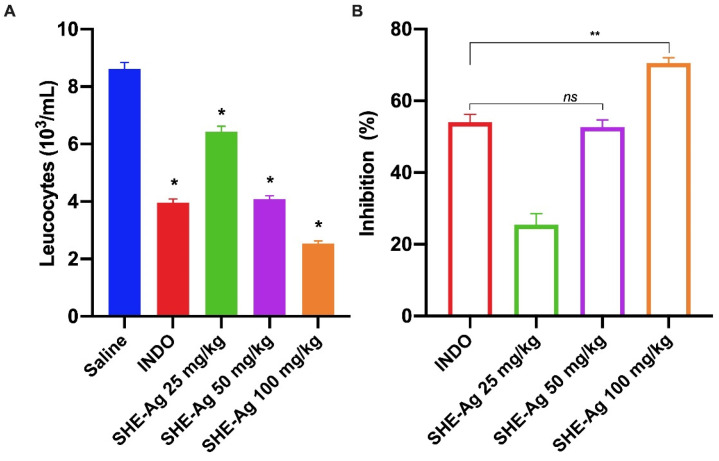
Effect of administration of Stem Hydroalcoholic Extract from *A. glaziovii* on leukocyte migration into the air pouch. (**A**) Total leukocytes and neutrophils 6 h post carrageenan administration. (**B**) Percentage of inhibition of leukocyte migration. All results were expressed as mean ± standard error of the mean (SEM). ns, non-significative. * *p* < 0.05, ** *p* < 0.01.

**Table 1 pharmaceutics-16-01298-t001:** Evolution of food and water consumption in the acute oral toxicity test.

		SHE-Ag (mg/kg)	
Parameter	Saline	2000	500
Water consumed (mL)	23.09 ± 1.50	19.07 ± 1.04 *	24.03 ± 1.86
Food consumed (g)	14.11 ± 0.55	10.96 ± 0.97 *	13.47 ± 0.83
Weight gain (g)	3.51 ± 0.31	3.62 ± 0.25	3.42 ± 0.33

SHE-Ag: Stem Hydroalcoholic Extract from *A. glaziovii*. * Significantly different (*p* < 0.05) from control (saline-treated mice). Statistical analysis was performed by analysis of variance (ANOVA) followed by Bonferroni’s test.

**Table 2 pharmaceutics-16-01298-t002:** Biochemical parameters of the blood of mice treated with SHE-Ag.

		SHE-Ag (mg/kg)	
Parameter	Saline	2000	500
ALB (g/dL)	20.14 ± 1.62	26.45 ± 1.84 *	20.76 ± 1.54
ALT (U/L)	59.36 ± 4.25	70.04 ± 5.18 *	60.25 ± 4.34
AST (U/L)	92.05 ± 7.18	115.77 ± 5.39 *	90.31 ± 4.90
TP (g/dL)	66.09 ± 5.13	68.29 ± 4.23	65.50 ± 5.68
ALP (IU/L)	11.14 ± 0.42	10.91 ± 0.31	10.77 ± 0.65
GGT (U/L)	10.26 ± 0.65	9.89 ± 0.53	9.71 ± 0.82
Urea (mg/dL)	0.32 ± 0.06	0.34 ± 0.05	0.36 ± 0.04
CREAT (mg/dL)	7.44 ± 0.59	7.26 ± 0.47	7.31 ± 0.40
TC (mg/dL)	91.03 ± 7.83	96.40 ± 8.26	90.53 ± 5.54
TG (mg/dL)	84.97 ± 5.90	87.64 ± 5.12	83.42 ± 6.65

ALB: Albumin; ALT: Alanine aminotransferase; AST: Aspartate transaminase; TP: Total protein; ALP: Alkaline phosphatase; GGT: gamma-glutamyl transferase; CREAT: Creatinine; TC: Total cholesterol; TG: Triglycerides. SHE-Ag: Stem Hydroalcoholic Extract from *A. glaziovii*. * Significantly different (*p* < 0.05).

**Table 3 pharmaceutics-16-01298-t003:** Hematological parameters of the blood of mice treated with SHE-Ag.

		SHE-Ag (mg/kg)	
Parameter	Saline	2000	500
Erythrocytes (10^6^/mm^3^)	6.21 ± 0.42	6.07 ± 0.58	6.59 ± 0.41
Hematocrit (%)	31.18 ± 2.63	27.68 ± 2.02 *	32.12 ± 2.55
Hemoglobin (g/dL)	13.37 ± 0.40	10.64 ± 0.37 *	13.27 ± 0.45
MCV (fL)	38.55 ± 2.95	40.13 ± 3.25	41.51 ± 4.18
MCH (pg)	15.79 ± 0.62	12.88 ± 0.84 *	15.10 ± 0.98
MCHC (%)	30.67 ± 2.43	31.02 ± 2.72	32.13 ± 3.06
Leukocytes (10^3^/mm^3^)	7.68 ± 0.56	7.33 ± 0.49	7.09 ± 0.52
Segmented (%)	63.46 ± 4.87	61.82 ± 3.63	59.66 ± 4.19
Lymphocytes (%)	24.55 ± 2.51	25.79 ± 2.49	26.12 ± 2.53
Monocytes (%)	3.42 ± 0.28	3.63 ± 0.31	3.28 ± 0.37

MCV: mean corpuscular volume; MCH: mean corpuscular hemoglobin; MCHC: mean corpuscular hemoglobin concentration. SHE-Ag: Stem Hydroalcoholic Extract from *A. glaziovii*. * Significantly different (*p* < 0.05).

**Table 4 pharmaceutics-16-01298-t004:** Determination of the number of micronucleated polychromatic erythrocytes (MPCE) in the peripheral blood of mice treated with SHE-Ag.

Treatments	Collection Time	Number of MNPCE per Animal					Mean MPCE
		M1	M2	M3	M4	M5	
Saline	24 h	1	1	0	0	1	0.60 ± 0.04
	48 h	0	1	1	0	1	0.60 ± 0.04
SHE-Ag (2000 mg/kg)	24 h	0	1	0	1	1	0.60 ± 0.04
	48 h	0	1	1	1	0	0.60 ± 0.04
CPA (50 mg/kg)	24 h	26	28	31	28	26	27.50 ± 2.42 *
	48 h	29	25	27	31	32	28.80 ± 2.30 *

M: mice. CPA: cyclophosphamide. SHE-Ag: Stem Hydroalcoholic Extract from *A. glaziovii*. * Significantly different from the control (saline-treated group) (*p* < 0.05).

## Data Availability

Data is contained within the article or Appendix A.

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
