# Peer review of "Apodanthera glaziovii (Cucurbitaceae) Shows Strong Anti-Inflammatory Activity in Murine Models of Acute Inflammation"

_pharmaceutics, 2024, doi:10.3390/pharmaceutics16101298_

Round 1
Reviewer 1 Report
Comments and Suggestions for Authors
Topic is interesting and worth of investigation.
Manuscript is well organized.
Graphical abstract is nice.
Results are presented in a fair way.
I suggest including more keywords that reflect the work presented in the paper.
References should be numbered in order of appearance and indicated by a numeral or numerals in square brackets-e.g., [1] or [2,3], or [4–6]. See the journal instructions for authors.
Abbreviations should be explained when first mentioned in the text.
Line numbers are not included in the text.
Manuscript needs technical improvement.
In several places looks like references are not included (Error! Reference source not found.). It should be checked and corrected.
Discussion is fine.
Figure 2. Diagram B is missing ns, non-significative, or * (p<0.05) for samples DEX and SHE-Ag 25 mg/kg. Are these samples significantly different or not?
Same comment applies for Figure 3 B and C, and Figure 4B.
I suggest including a brief conclusion remarks.
Comments on the Quality of English Language
English could be checked, but no issues detected.
Author Response
Reviewer #1
- Topic is interesting and worth of investigation.
- Manuscript is well organized.
- Graphical abstract is nice.
- Results are presented in a fair way.
Response: We appreciate the comments about the manuscript.
- I suggest including more keywords that reflect the work presented in the paper.
Response: We appreciate the suggestion. Keywords number were adjusted in the revised manuscript.
- References should be numbered in order of appearance and indicated by a numeral or numerals in square brackets-e.g., [1] or [2,3], or [4–6]. See the journal instructions for authors.
Response: We appreciate the revision. All references have been reviewed and adjusted accordingly.
- Abbreviations should be explained when first mentioned in the text.
Response: We appreciate the correction. The manuscript was revised and abbreviations are now explained.
- Line numbers are not included in the text.
Response: We appreciate the comment. The lines number has been inserted into the text.
- Manuscript needs technical improvement.
Response: We appreciate the suggestion. The manuscript has been reviewed and now, presented with significant improvements.
- In several places looks like references are not included (Error! Reference source not found.). It should be checked and corrected.
Response: We appreciate the comment. All references have been reviewed and the file was corrected.
- Discussion is fine.
Response: We appreciate the comment.
- Figure 2. Diagram B is missing ns, non-significative, or * (p<0.05) for samples DEX and SHE-Ag 25 mg/kg. Are these samples significantly different or not?
Response: We appreciate the comment. The figure has been modified as suggested. The DEX and SHE-Ag 25mg/Kg groups are not significantly (ns) different.
- Same comment applies for Figure 3 B and C, and Figure 4B.
Response: We appreciate the comment. The figure has been adjusted.
- I suggest including a brief conclusion remarks.
Response: The conclusion is already in the manuscript. Thanks for suggestion.

Reviewer 2 Report
Comments and Suggestions for Authors
In the manuscript entitled “Stem hydroalcoholic extract from Apodanthera glaziovii (Cucurbitaceae), a species endemic to the Brazilian semi-arid region (Caatinga), shows strong anti-inflammatory activity in murine models of acute inflammation” by de Oliveira Andrade and colleagues, the authors evaluate the metabolites present from A. glaziovii through qualitative analyses using high-performance liquid chromatography (HPLC) and 1H and 13C nuclear magnetic resonance (NMR). In addition, the study examined its toxicity both in vivo and in vitro, as well as evaluating its anti-inflammatory activity in in vivo models.
The work is well written and the experiments are well thought out and well done, some experiments need to be increased.
Minor point
- In section 3.1 the Authors in order to determine the exact concentration of saponins in SHE-Ag and confirm the results of the qualitative foam test should quantitatively analyze the saponin content. This can be done using methods such as gravimetric analysis, spectrophotometric methods or HPLC.
- In section 3.2, in order to confirm the molecular structures of triterpenoid saponins and other components and identify any other minor compounds present in the extract, the Authors should perform mass spectrometry, such as electrospray ionization mass spectrometry (ESI-MS) or matrix-assisted laser desorption ionization mass spectrometry, to accurately determine the molecular weights of the compounds corresponding to the peaks observed in the HPLC chromatogram.
- To evaluate the general cytotoxicity of SHE-Ag on various cell types and assess its safety for potential therapeutic use, the Authors, in section 3.3, should conduct cytotoxicity tests using different cell lines, such as human skin fibroblasts, liver cells (hepatocytes) or immune cells (macrophages). Furthermore, in order to investigate whether SHE-Ag affects cell membranes without causing complete lysis (which would provide information about its mechanism of interaction with cell membranes), the authors should use assays such as lactate dehydrogenase (LDH) release or propidium iodide (PI) staining to assess membrane integrity and permeability in SHE-Ag-treated cells. Finally, to determine whether SHE-Ag induces oxidative stress or possesses antioxidant properties that could protect cells from oxidative damage, the authors should measure the levels of reactive oxygen/nitrogen species (ROS/RNS) and antioxidant enzyme activities in cells exposed to SHE-Ag.
- To determine the specific effects of SHE-Ag on inflammatory mediators (that would help elucidate its anti-inflammatory mechanism) the authors could analyze the levels of key pro- and anti-inflammatory cytokines (e.g., TNF-α, IL-6, IL-10) and chemokines in air sac exudate or blood serum after SHE-Ag treatment.
Author Response
Reviewer #2:
- In section 3.1 the Authors in order to determine the exact concentration of saponins in SHE-Ag and confirm the results of the qualitative foam test should quantitatively analyze the saponin content. This can be done using methods such as gravimetric analysis, spectrophotometric methods or HPLC.
Response: We appreciate the comment. The objective was to perform a preliminary qualitative analysis using traditional techniques. Saponins are secondary metabolites of plants that usually have a large chemical diversity, usually found as a series of related molecules, and their quantification by methodologies such as HPLC-DAD is challenging. Furthermore, tests based on colorimetric reactions (using vanillin) are known to present false positive results due to interaction with other metabolites found in complex matrices. Thus, the methods applied in our study were qualitative, as well as NMR analysis for the verification of saponins and qualitative-quantitative methods for the quantification of other SHE-Ag metabolites.
- In section 3.2, in order to confirm the molecular structures of triterpenoid saponins and other components and identify any other minor compounds present in the extract, the Authors should perform mass spectrometry, such as electrospray ionization mass spectrometry (ESI-MS) or matrix-assisted laser desorption ionization mass spectrometry, to accurately determine the molecular weights of the compounds corresponding to the peaks observed in the HPLC chromatogram.
Response: We appreciate the comments and agree with the suggestions. However, in this work we performed a phytochemical study with classical methods, as well as using modern methodologies such as NMR and HPLC-DAD. From these methodologies, it was possible to confirm the presence of compounds characteristic of the genus Apodanthera and for the first time of the species A. glaziovii. In the NMR analyses it was possible to identify the presence of characteristic and non-overlapping signals of triterpene saponins found in the Cucurbitaceae family, as well as the absence (or very low concentrations) of phenolic compounds. As previously stated, this is the first report describing a phytochemical analysis for this species, as well as the pharmacological activity and safety of use.
- To evaluate the general cytotoxicity of SHE-Ag on various cell types and assess its safety for potential therapeutic use, the Authors, in section 3.3, should conduct cytotoxicity tests using different cell lines, such as human skin fibroblasts, liver cells (hepatocytes) or immune cells (macrophages). Furthermore, in order to investigate whether SHE-Ag affects cell membranes without causing complete lysis (which would provide information about its mechanism of interaction with cell membranes), the authors should use assays such as lactate dehydrogenase (LDH) release or propidium iodide (PI) staining to assess membrane integrity and permeability in SHE-Ag-treated cells. Finally, to determine whether SHE-Ag induces oxidative stress or possesses antioxidant properties that could protect cells from oxidative damage, the authors should measure the levels of reactive oxygen/nitrogen species (ROS/RNS) and antioxidant enzyme activities in cells exposed to SHE-Ag.
Response: We appreciate the comments and suggestions. However, we performed preliminary tests using erythrocytes to evaluate the interaction of SHE-Ag with biological membranes (session 3.3). Thus, with the absence of hemolytic effect, acute oral toxicity tests were performed in mice, which is a highly robust test with interaction systemic. Regarding the investigation of ROS/RNS levels, this was a pioneering study in which we aimed to evaluate the anti-inflammatory activity and safety of use. Our group has been developing subsequent tests with different approaches including analysis of mechanisms of action.
- To determine the specific effects of SHE-Ag on inflammatory mediators (that would help elucidate its anti-inflammatory mechanism) the authors could analyze the levels of key pro- and anti-inflammatory cytokines (e.g., TNF-α, IL-6, IL-10) and chemokines in air sac exudate or blood serum after SHE-Ag treatment.
Response: We appreciate the comments. TNF-a and IL-1b levels were quantified in the carrageenan-induced peritonitis model, showing that treatment with SHE-Ag promoted a strong reduction of these pro-inflammatory cytokines. As previously stated, our group has been working to elucidate the mechanisms of action in the anti-inflammatory response of triterpene saponins from A. glaziovii.
